# Quantifying the Impact of Chronic Obstructive Sialadenitis on Quality of Life

**DOI:** 10.3390/jcm14217560

**Published:** 2025-10-24

**Authors:** Alvaro Sánchez Barrueco, Gonzalo Díaz Tapia, Ignacio Alcalá Rueda, William Aragonés Sanzen-Baker, Jessica Mireya Santillán Coello, Pilar Benavent Marín, Alberto Valentín González, Ignacio Mahillo Fernández, Carlos Cenjor Español, José Miguel Villacampa Aubá

**Affiliations:** 1Medicine Faculty, Universidad Alfonso X el Sabio (UAX), Avenida de la Universidad 1, 28691 Villanueva de la Cañada, Madrid, Spain; gdiazt@quironsalud.es (G.D.T.); dr.alcalarueda.orl@gmail.com (I.A.R.); william.aragones@quironsalud.es (W.A.S.-B.); jessica.santillan@quironsalud.es (J.M.S.C.); avg1994valentin@gmail.com (A.V.G.); 2ENT and Cervicofacial Surgery Department, Fundación Jiménez Díaz University Hospital, Avenida de los Reyes Católicos, 2, 28040 Madrid, Spain; benaventmarinp@gmail.com (P.B.M.); carlos.cenjor@gmail.com (C.C.E.); jmvillacampa@fjd.es (J.M.V.A.); 3ENT and Cervicofacial Surgery Department, General Villalba University Hospital, M-608 Carretera, M-608, Km 41, 28400 Collado Villalba, Madrid, Spain; 4Biostatistics and Epidemiology Unit, Health Research Institute, Fundación Jiménez Díaz University Hospital, Avenida de los Reyes Católicos, 2, 28040 Madrid, Spain; imahillo@fjd.es; 5Medicine Faculty, Universidad Autónoma de Madrid, Calle Arzobispo Morcillo, 4, 28029 Madrid, Spain

**Keywords:** quality-of-life, sialadenitis, chronic obstructive sialadenitis, sialolithiasis, sialendoscopy

## Abstract

**Objectives**: To evaluate the loss of quality of life (QoL) in patients with chronic obstructive sialadenitis (COS) using the Chronic Obstructive Sialadenitis Questionnaire (COSQ). **Methods**: The COSQ was administered to patients diagnosed with COS, with the diagnosis confirmed by sialendoscopy. Epidemiological data, obstructive causes and potentially obstructive entities were collected. QoL was assessed using the COSQ. **Results**: A total of 344 glands in 278 patients with COS were analyzed. Most patients were women (71.94%), and the main obstructive cause was stenosis (47.96%), followed by lithiasis, lack of papilla distensibility (LPD), and mucus plug. Stenosis was significantly more frequent in the parotid gland and in women, whereas lithiasis predominated in the submandibular gland and in men. The mean COSQ score was 30.55 and it was significantly higher in women (*p* < 0.005), parotid gland (*p* < 0.005), and in long-standing cases (*p* < 0.05). Stenosis and LPD were the obstructive causes with the greatest impact on QoL (*p* < 0.005), while lithiasis had the least impact. Potentially Obstructive Entities (POEs), such as eosinophilic sialodochitis, Sjögren’s syndrome, or radioiodine-induced sialadenitis, were associated with a notable loss of QoL. Likewise, patients without associated POEs presented significantly lower COSQ values (*p* < 0.05). **Conclusions**: COS significantly affects QoL, particularly in women and in cases of parotid gland, stenosis, and LPD. Lithiasis has the least impact on QoL. It is important to standardize a thorough evaluation of COS using validated tools such as the COSQ, which are fundamental for understanding the disease and predicting the outcomes of therapeutic interventions.

## 1. Introduction

Chronic obstructive sialadenitis (COS) is a benign condition resulting from the obstruction of the salivary glands, either in the main duct or in various branches of the ductal system. The most common causes of obstruction include lithiasis [1] and stenosis [2,3,4], as well as other factors such as mucus plugs or lack of papilla distensibility (LPD) [2]. These obstructive causes can occur idiopathically or be associated with diseases that present with obstructive glandular manifestations, such as juvenile recurrent parotitis (JRP), radioiodine-induced sialadenitis (RIS), Sjögren’s syndrome (SS), or eosinophilic sialodochitis (ES) [5], in addition to extrinsic factors that can compress the ductal system. 

The initial diagnosis is clinical and supported by imaging techniques such as ultrasound, computed tomography (CT), and magnetic resonance sialography (MR-Si) [6,7]. According to recent evidence, ultrasound should be considered the first-line diagnostic tool due to its high sensitivity (>90%) in obstructive salivary gland diseases, its non-invasive nature, and its ability to be used dynamically and repeatedly [7]. While CT shows limited sensitivity for non-lithiasic causes and involves radiation, MR-Si offers detailed anatomical information and high diagnostic accuracy, particularly in complex or non-obstructive inflammatory cases [2]. Therefore, depending on available resources, MR-Si can serve as a complementary tool when ultrasound findings are inconclusive. Sialendoscopy might become the gold standard for obstructive salivary gland diseases, as suggested by literature [8], especially for the direct assessment of intraductal pathology, but its combination with ultrasound has been shown to enhance diagnostic precision and clinical decision making.

In recent decades, there has been a relentless effort to evaluate the impact on quality of life (QoL) and outcomes following healthcare interventions, both medical and surgical. Patient-reported outcome measures allow the evaluation of the results of an intervention from the patient’s perspective. To facilitate the study of QoL loss associated with COS (QoL-COS), various tools have been developed, ranging from generic quality of life questionnaires (6–9) to specific questionnaires. Among the latter are the MGSS [9], MSSS [10], S-OHIP [11], COSQ [12], COSS [13], or SPIT [14]. However, only SPIT and COSQ ensure sensitivity to change, allowing a thorough evaluation of the pre- and post-sialendoscopy impact. The COSQ is a specific questionnaire consisting of 18 questions, with scores ranging from 0 to 4; it demonstrates high feasibility, stability, internal consistency, and high responsiveness to change [12] (Appendix A). 

It is well established that COS causes a decrease in the QoL of patients, often underestimated. Currently, the best part of published studies focuses on improving the QoL of patients following minimally invasive interventions for salivary obstruction. However, there is no detailed study of QoL-COS using validated methods in all aspects. 

Our objective is to explore, through a cross-sectional analysis, the clinical and obstructive factors associated with the greater impairment of QoL-COS, using a previously validated questionnaire tailored to this clinical context.

## 2. Materials and Methods

### 2.1. Study Design and Ethical Considerations

The study was approved by the Fundación Jiménez Díaz University Hospital Ethics Committee (FJD-SAC-16-01). All patients were informed of the study and signed a specific informed consent form permitting subsequent analysis of their data. 

### 2.2. Patient Selection and Questionnaire Administration

From January 2017 to June 2024, the COSQ was administered to patients who attended the salivary gland pathology clinic with symptoms compatible with COS, and who subsequently underwent sialendoscopy at the University Hospital Fundación Jiménez Díaz and the University Hospital General de Villalba. Exclusion criteria were: (i) age <16 years; (ii) inability to complete the questionnaire; (iii) purely intraparenchymal stones not amenable to endoscopic management; and (iv) known salivary gland neoplasia. No patients declined to participate in the study.

In patients with a single affected gland, the questionnaire was administered once. For those patients with multiple affected glands, one separate questionnaire was completed for each affected gland, considering that each gland may contribute differently to the patient’s overall quality of life. This allowed us to assess the impact of each individual gland on QoL-COS in a specific and independent manner.

### 2.3. Diagnosis

The initial diagnosis was predominantly made using MR-Si, while the definitive diagnosis was confirmed intraoperatively by sialendoscopy, particularly in cases of LPD and mucus plug, where radiological evaluation is limited. 

The procedure associated with sialendoscopy was selected based on the obstructive pathology and the feasibility of extraction, including purely endoscopic approaches or combined endobuccal or transfacial approaches, in accordance with previous algorithms [15]. Representative intraoperative sialendoscopic findings are shown in Figure 1.

### 2.4. Epidemiological and Salivary Obstruction Data

Epidemiological data were collected, including age, sex, affected gland, and duration of COS (years since the first symptom). The recorded obstructive causes included stenosis, lithiasis, LPD, and mucus plug. Additionally, possible associations with diseases presenting obstructive glandular manifestations, referred to as Potentially Obstructive Entities (POEs) such as RIS, SS, JRP, and ES, as well as extrinsic factors with the potential to compress the salivary duct, were specified.

Stenoses were stratified based on the LSD classification [16] into S1 (single stenosis starting from the first bifurcation), S2 (single ductal stenosis), S3 (panductal stenosis), and S4 (stenosis involving both the main duct and intraglandular ducts). It was noted whether the stenoses were single or combined with various grades. 

Lithiasis were classified as ductal or hilar. Purely intraparenchymal lithiasis were excluded due to the limitations in resolving them endoscopically and their typically minimal impact on QoL. These cases were identified and excluded based on preoperative radiological imaging, primarily MR-Si. The size of the stones was recorded once they were extracted. In cases where multiple stones were found, their impact was analyzed by adding the size of all the stones found. The LSD classification [16] was not applied to lithiasis, given the potential for conversion between stages and the lack of information about their location.

LPD was diagnosed in cases where it was manifestly difficult to achieve dilation of the papilla, which was non-elastic and fibrotic, regardless of its morphology or position [2]. This obstructive factor has distinctive epidemiological features [2]; similarly, it represents a condition not previously contemplated in the LSD classification [16] and not related to anatomical variations. 

Mucus plugs were diagnosed when there was evidence of an independent accumulation of mucus not adjacent to another obstructive cause, and their dispersion with saline solution was impaired. 

Dilations were not considered a primary obstructive problem but rather a consequence, and thus they were analyzed independently. Generally, they were diagnosed using MR-Si, confirmed endoscopically, and classified according to the LSD classification into D1 (single dilation), D2 (multiple dilations), and D3 (generalized dilation). Dilations resulting from stimulation with sialagogues during MR-Si were not considered.

### 2.5. Statistical Analysis

Statistical analyses were performed using Python v3.13.7. Due to non-normal data distribution verified by histogram and Shapiro–Wilk test, median and range values were utilized for descriptive statistics. Categorical data were analyzed using Chi-square or Fisher’s exact tests when appropriate, with odds ratios (ORs) calculated for epidemiological associations. COSQ scores were compared using Mann–Whitney U tests for binary variables and Kruskal–Wallis tests for variables with three or more categories. An independent *t*-test compared COSQ scores between single and multi-gland patients to validate individual gland assessment. Spearman’s test was employed for correlations between continuous variables, and ANOVA analyzed sex and symptom duration against obstructive factors and POEs. Dunn’s post hoc test with Bonferroni correction assessed COSQ scores across obstructive factors.

To evaluate potential bias from intra-patient correlation due to gland-level analysis, the intraclass correlation coefficient (ICC) was calculated using a one-way random, absolute agreement model (ICC(1,1)). The ICC was 0.982 (95% CI: 0.971–0.992), confirming strong within-patient similarity and highlighting the need to correct for this clustering. Thus, a linear mixed-effects model with random patient intercepts was applied using Python (statsmodels package), controlling patient-level variance. This approach preserves the individual gland data while preventing inflation of type I error.

## 3. Results

Applying the established selection criteria, 344 glands were analyzed in 278 patients. Most patients had only one gland treated (77.33%, n = 215), while others had two glands treated synchronously (21.94%, n = 61). One patient had three glands treated, and another had four glands treated.

### 3.1. Epidemiological Data

The study population showed a female-to-male ratio of 2.56:1. The mean age at which patients were treated was 49.67 years (SD 14.12 years), with a median duration of COS symptoms of 2.15 years (range 0.2–9). An equal number of parotid glands and submandibular glands were treated, 172 each. All epidemiological values are presented in Table 1.

### 3.2. Obstructive Factors and POEs

Overall, the most important obstructive factors found were stenosis (47.96%, n = 165) and lithiasis (32.27%, n = 111), followed by LPD and mucus plug. These obstructive findings were present individually in 92.73% of the operated salivary glands (n = 319) or due to a combination of several factors in the salivary gland (7.27%, n = 25). All data is provided in Appendix A.

Stenosis occurred significantly more frequently in the parotid gland (*p* < 0.005, OR 8.7) and in women (*p* < 0.005, OR 2.3), both globally and stratified by the degree of stenosis (*p* < 0.005). Conversely, neither the presence nor the degree of stenosis was influenced by the patient’s age or duration of symptoms. The most common degree of strictures was S2 (49.69%, n = 82), indicating a stenosis of a segment of the main salivary duct. 

Conversely, lithiasis occurred more frequently in the submandibular gland (*p* < 0.005, OR 5.12) and in men (*p* < 0.005, OR 3.94). Regarding size, a significant difference was observed, being considerably larger in men (7 mm [range 2–23]) than in women (4.8 mm [range 0.8–13]) (*p* < 0.005). Additionally, lithiasis was present in younger patients (*p* = 0.026) and those with a shorter duration of symptoms (*p* = 0.007). The characteristics of submandibular hilar lithiasis were studied independently, without showing any predominance by sex, age, or duration of symptoms. Similarly, the size of the stone was not influenced by the affected gland, age, sex, or duration of symptoms.

LPD occurred more often in older individuals (53.5 years [range 17–80], *p* = 0.048), predominantly in women (*p* = 0.031, OR 2.28), and in the submandibular gland (*p* < 0.005, OR 6.43). Conversely, LPD was not influenced by the duration of symptoms.

Mucus plug occurred significantly more often in younger individuals (48 years [range 21–69], *p* = 0.03) and with a longer duration of symptoms (*p* < 0.005). They predominantly affected the parotid gland (*p* < 0.001, OR 5.84). Conversely, mucus plugs were not influenced by sex.

The studied POEs (RIS, SS, JRP, or ES) were not influenced by sex or duration of symptoms. However, significantly, the lowest median age was seen in JRP (16 years) and the highest in RIS (54.5 years). The parotid gland was the most significantly affected in RIS (*p* = 0.0053) and SS (*p* < 0.005). No relationship was found between POEs and the different obstructive factors. The obstructive causes leading to COS, whether in their idiopathic forms or related to POEs, are summarized in Figure 2.

The study of dilation revealed that it was more frequent in older individuals (*p* < 0.005) and in the parotid gland (77.42%, *p* < 0.005, OR 3.75), with no relation to the duration of symptoms or sex. Dilation was statistically related to the obstructive cause that provoked it, specifically stenosis (OR 18.85).

Extrinsic factors that could compress the salivary duct included scars from sublingual ranula excision (n = 3), facial scars from a dog bite (n = 1), and excessive amounts of hyaluronic acid injected into the face (n = 2).

### 3.3. COSQ Values and Epidemiology

The high degree of within-patient consistency in COSQ scores, combined with the statistical modelling described above, supports evaluating each gland as a separate observational unit. First, the intraclass correlation coefficient (ICC [1,1]) was 0.982 (95% CI 0.971–0.992), indicating that 98% of the total variance in COSQ scores resides between patients rather than between glands within the same patient. Variance-component estimates from the mixed-effects model quantified this further: σ^2^_patient = 161.3 versus σ^2^_residual = 14.8, a 10.9:1 ratio, confirming that individual glands contribute independent information once patient clustering is accounted for. Second, a likelihood-ratio test comparing the random-intercept mixed model against a simple fixed-effects model yielded χ^2^(1) = 112.7, *p* < 0.0001, demonstrating that the inclusion of the patient-level random effect significantly improves model fit. Finally, the gland-level analysis maintained sensitivity and avoided bias: mean COSQ in single-gland cases (n = 215) was 27.19 (±13.11 SD) versus 35.95 (±14.38 SD) in multi-gland cases (n = 132; *p* < 0.0001, by mixed-model Wald test), and effect estimates for sex, obstruction type, and gland site remained consistent in direction and magnitude whether calculated per-patient or per-gland (Appendix A). Additionally, glands from patients with multi-gland involvement (affecting two, three, or four glands) consistently showed higher COSQ scores compared to glands from patients with single-gland involvement, reflecting a greater burden of disease. Across all groups, epidemiological, clinical and obstructive associations were consistent, confirming that per-gland COSQ analysis remains unbiased, sensitive and accurate (even in multigland cases). Appendix A provide the entire comparative analysis across these groups. Taken together, these findings justify the use of COSQ at the gland level, provided that patient-level clustering is modelled via random effects. 

The mean COSQ score was 30.55 (SD 14.24).

Among the significantly influential factors was sex, with women reflecting higher COSQ values (average of 5.56 points higher) (*p* < 0.005). The duration of symptoms had a proportional influence on COSQ values, with longer symptom durations corresponding to higher COSQ scores (*p* = 0.01297). Additionally, quality of life was found to be significantly more affected in the parotid gland (*p* < 0.005) (Figure 3). 

No significant correlation was found between age and COSQ score. 

All factors are presented in Table 2.

### 3.4. COSQ Values and Obstructive Factor

The COSQ values were significantly different depending on the obstructive pathology (*p* < 0.005). Excluding obstructive causes that accounted for less than 1% of the sample, the highest median COSQ was found in Stenosis + LPD (COSQ 37.5), Stenosis (COSQ 33), and LPD (COSQ 31) (Figure 4). Overall, parotid COS had a higher median COSQ (35 vs. 26) (*p* < 0.005).

Supported by the post hoc analysis, lithiasis was found to be the pathology with the least impact on COSQ, with its values being significantly lower than stenosis (*p* < 0.001) and LPD (*p* = 0.015). Additionally, there was no difference in COSQ values based on the location of the lithiasis in the salivary duct, the affected gland, the size of the lithiasis, or the number of lithiasis causing COS.

Stenosis was overall one of the most symptomatic obstructive causes. Furthermore, the degree of stenosis was a significantly influential factor on COSQ values, being notably higher in S4 (*p* < 0.005), which indicates stenoses that affected the entire salivary ductal tree. No significant differences in COSQ scores were observed when analyzing different combinations or grades of stenosis, or based on whether the stenosis affected the parotid or submandibular gland.

In LPD, COSQ scores were significantly higher in parotid than in submandibular glands (medians 38.5 [20–55] vs. 30.5 [7–62]; *p* = 0.016).

Dilation did not significantly influence COSQ values, both globally and stratified by degrees. Moreover, the gland affected by dilation did not significantly impact COSQ values.

Patients without associated POEs presented significantly lower COSQ values (*p* = 0.03431). ES was the most symptomatic POE (COSQ = 65, *p* = 0.0482), followed by SS (COSQ = 42, *p* = 0.8) and RIS (COSQ = 36, *p* = 0.21). No statistically significant differences in COSQ values were found based on the gland affected by the POE. 

All COSQ values are presented in Table 3 and Table 4.

## 4. Discussion

COS presents specific epidemiological characteristics that must be studied independently. Additionally, the loss of QoL-COS should be highlighted, stratifying this impact by the factors that directly affect COS. Our series, with 344 glands analyzed, is the largest to date studying QoL-COS using a tool that has successfully completed all validation stages for use as a quality-of-life instrument [12], along with a detailed analysis of the epidemiological factors surrounding COS. 

Our findings confirm that the COSQ is a valid tool for assessing obstructive symptom burden that can be applied independently to each affected gland. The significant differences observed between single-gland and multi-gland patients indicate that, although many items in the COSQ refer to general symptoms, the overall score reflects the local contribution of each gland to the patient’s overall well-being. Moreover, in patients with multiglandular involvement, it is not uncommon for symptoms to be asymmetrical, leading them to assign different scores when the questionnaire is administered separately for each gland. Our mixed-effects model already corrects for patient-level clustering, and the consistency of results in the single-gland subgroup supports the robustness of our findings. This justifies the use of individual gland scoring that allows researchers and clinicians to better understand the distinct contribution of each gland to the patient’s overall burden of disease [4]. 

According to the COSQ data, factors influencing QoL-COS were demonstrated to include sex, duration of symptoms, the affected gland, and the obstructive factor causing COS.

According to our series, COS predominantly affects women, reaching 71.94%, slightly above other published series that report around 65% [2,9,10,14,17]. Stenosis is the primary obstructive cause (47.96%), above lithiasis (32.27%), LPD, and mucus plugs. This finding, contrary to the general perception among non-specialized medical practitioners, is a constant in publications of large series [2,4,10,17], although other series only characterize it as lithiasic or non-lithiasic [13] or analyze obstructive factors and associated diseases at the same level [18]. This can introduce bias in the analysis of these studies, reducing the actual prevalence of stenosis. Therefore, it is important to attempt to analyze obstructive factors independently, whether they are idiopathic forms or related to a POE. Other obstructive factors include LPD [2] and mucus plugs, which account for about a quarter of the sample, with specific epidemiological characteristics. Similarly, the fact that the same patient may suffer from a combination of several obstructive factors has a significant impact on QoL-COS.

Stenosis is more frequent in the parotid gland and in women, as previously reported [2,11,14,19]. Conversely, lithiasis is more frequent in the submandibular gland [11] and in men, who present significantly larger stones. This finding contradicts previous postulates where lithiasis was not considered gender-dependent [2,14,20]. LPD, confirming previous publications [2], occurs in older individuals, predominantly in women and in the submandibular gland. Although LPD is not currently considered a standard obstructive pattern, our findings suggest it may represent a relevant obstructive mechanism and should be increasingly taken into account. The importance of the papilla in salivary obstruction is highlighted in the recent literature. Aničin et al. [21] proposed a classification of the Wharton’s duct papilla and identified a nearly closed papilla (Type D) that makes endoscopic cannulation difficult, a concept that aligns with the LPD described previously [2] and in the present study. Similarly, Koch et al. [22] reported that nearly 60% of submandibular duct stenosis occur at the papilla and defined a subtype caused by anatomic duct narrowness, reinforcing that a non-distensible papilla (LPD) is a key obstructive factor with significant clinical impact on salivary gland function and patient quality of life. However, LPD epidemiological differences and its clear resolution through sialendoscopy make it an obstructive factor among the already known ones, such as stenosis, lithiasis, and mucous plugs.

By other hand, mucus plugs are characteristic of younger individuals and predominantly affect the parotid gland. In our series, POEs predominantly involved the parotid gland, particularly in cases of RIS and SS. By contrast, no correlation was found between POEs and the specific obstructive factors, unlike in previous publications where stenosis was associated with RIS [23] or SS [24].

In recent years, several specific questionnaires have been developed to study the QoL-COS and, primarily, to demonstrate the improvement of COS symptoms after sialendoscopy. Among these, there are specific questionnaires exclusively for obstructive symptoms or xerostomia, such as the MGSS [9], or those that evaluate patient satisfaction after sialendoscopy, such as the COSS [13] or its standardized and validated version, SPIT [14]. Other questionnaires include the MSSS [10], S-OHIP [11], or COSQ [12]. However, only some of them ensure sensitivity to change, allowing for a comprehensive evaluation of the pre- and post-sialendoscopy impact. They often fail in structural validation, criterion validity, or construct validity. The MSSS is prospective and does not consider cases without stenosis or lithiasis, which account for up to 2.5% of procedures [10], likely related to undiagnosed LPD [2]. Additionally, some authors [9,25] grouped strictures into a global category, including idiopathic stenoses and related to pathologies (such as IgG4-related disease, RIS, or JRP), likely causing data dispersion and underestimating the true prevalence of stenosis. Some authors considered multiple gland involvement [26], contrary to our study, where a questionnaire was applied for each affected gland. Among all of them, the questionnaires that meet all the validation processes are SPIT and COSQ. Therefore, in our series, we used COSQ, following the previously applied methodology in submandibular hilar lithiasis [27].

The COSQ evaluates the impact of QoL-COS with higher values corresponding to greater loss of QoL. In our series, the median COSQ score was 30.55 (range 0–72). Novel findings showed that COSQ was influenced by sex and the duration of symptoms. Women presented higher COSQ values, and as the duration of symptoms increased, COSQ scores also increased. Additionally, the gland that was significantly associated with the greatest impact on QoL-COS was the parotid gland, contrary to some series [26]. The COSQ was influenced by the obstructive factor, with the greatest impact on QoL-COS associated with stenosis, whether alone or combined with LPD. Additionally, the degree of stenosis was shown to be a significant influencing factor, with greater impact in S4 stenoses. The presence of multiple stenoses, according to some authors, negatively influences QoL-COS [4], a finding not evidenced in our series. Furthermore, COSQ values for stenoses were not gland-dependent, comparable to the findings published by Delagnes et al. [26]. 

Consistent with previous publications [24], dilation had a significant relationship with stenosis (OR 18.85). However, dilation did not independently influence COSQ values. It is noteworthy that LPD [2] is one of the most symptomatic obstructive factors. Therefore, it should be considered an independent obstructive factor, beyond lithiasis, stenosis, or mucus plug, with distinct epidemiological characteristics.

Conversely, lithiasis was found to be the pathology with the least impact on COSQ, a finding previously reflected in other series [11,13,14,25]. There was no impact based on the location of the lithiasis, the gland, the size, or the number of stones, in accordance with other series [17].

The absence of associated POE resulted in significantly lower COSQ values. Among POEs, ES was the most symptomatic, followed by SS and RIS. This is in line with other series [4,13] reporting greater QoL-COS impairment in patients with SS. It is also plausible that symptoms commonly associated with certain POEs, such as xerostomia, may contribute to higher COSQ scores in some patients.

### 4.1. Limitations

The main limitation of our study is that the epidemiological data were obtained from a sample of patients who attended the salivary gland pathology clinic. Therefore, there may be a bias in extrapolating these findings to the general population prevalence, consistent with previous studies. We relied on the COSQ as the sole disease-specific PROM; generic health-status instruments (e.g., OHIP-14, EQ-5D) were not collected, and minimal clinically important difference for COSQ was not established. 

The questionnaire is currently available only in Spanish, although cross-cultural adaptation is underway; in order to allow its use in a more heterogeneous population. Patients < 16 years were excluded, which explains the low representation of JRP cases.

### 4.2. Clinical Implications and Future Directions

The availability of validated tools, including the COSQ and other instruments developed in recent years, makes it increasingly feasible to measure and monitor the quality of life in patients with COS. This confirms that QoL-COS is influenced by factors from both an epidemiological perspective and the associated obstructive cause or potentially obstructive entity. 

Therefore, COSQ should be incorporated into the routine management of patients with COS as a per-gland baseline and follow-up instrument. Practically, we suggest administering COSQ at the first visit and before any intervention for each symptomatic gland, then at standardized intervals during follow-up to support triage, shared decision-making, and longitudinal monitoring. Importantly, COSQ will very likely capture clinically meaningful changes in COS-QoL across the obstructive factors and POEs; enabling phenotype-informed and personalized strategies. 

Future multicenter work should define actionable thresholds and the minimal clinically important difference for COSQ, and prospectively test whether COSQ-guided pathways improve time-to-symptom control and patient-reported outcomes.

## 5. Conclusions

COS presents specific epidemiological characteristics that need to be studied independently. The loss of QoL-COS is evident and should be analyzed in detail.

In our series, we conclude that QoL-COS is significantly affected by both demographic factors (sex) and clinical factors (duration of symptoms, affected salivary gland, cause of obstruction, and associated POEs). In our cohort, women are more affected by COS and experience greater loss of QoL. Stenosis, both alone and associated with LPD, is the obstructive cause with the greatest negative impact on QoL-COS, followed by systemic diseases (SS, RIS). Lithiasis is the obstructive cause with the least impact on QoL-COS.

## Figures and Tables

**Figure 1 jcm-14-07560-f001:**
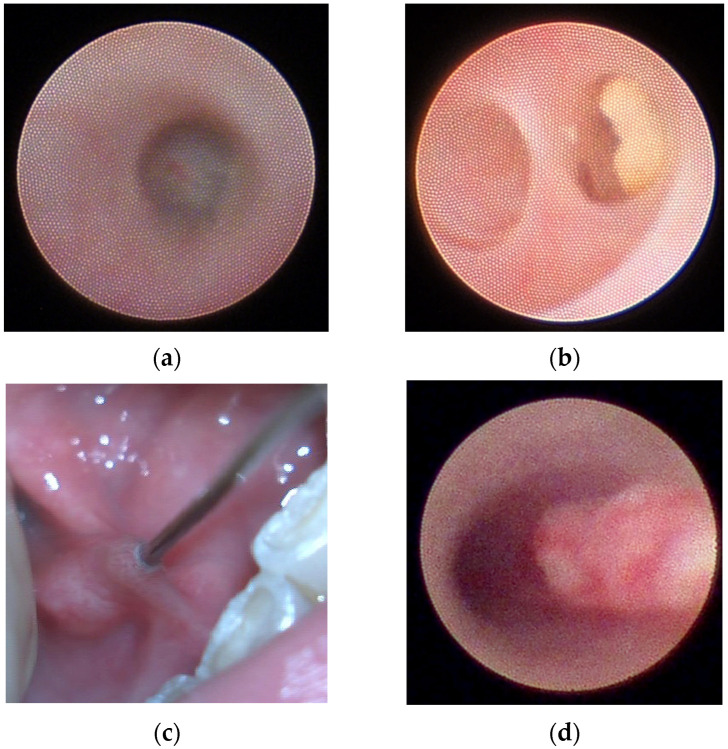
Representative intraoperative sialendoscopic findings. (**a**) Parotid S2 segmental stenosis with membranous translucid wall. (**b**) Second branch lithiasis. (**c**) Submandibular duct lack of papilla distensibility (LPD), a whitish ring surrounding the 00 dilator can be identified, markedly hindering dilation of the papilla. (**d**) Mucus plug filling the entire duct in a case of eosinophilic sialodochitis (ES).

**Figure 2 jcm-14-07560-f002:**
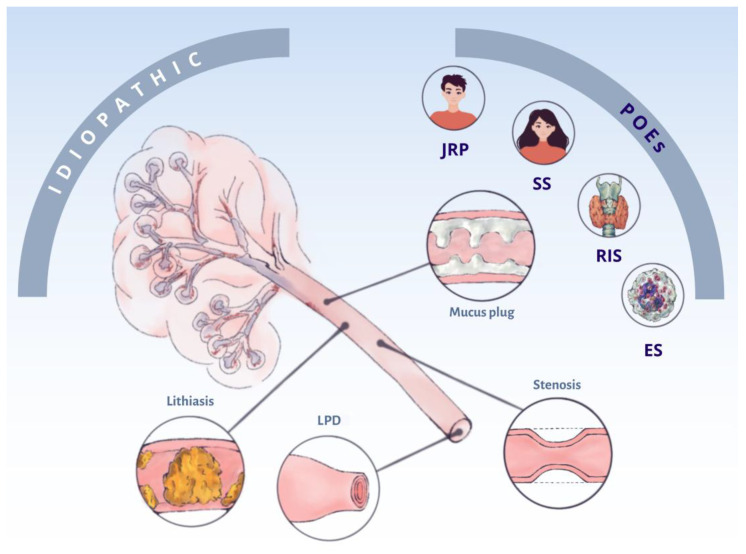
Conceptual map of the obstructive factors of chronic obstructive sialadenitis (COS) whether idiopathic or associated with potentially obstructive entities (POEs). LPD: lack of papilla distensibility; JRP: juvenile recurrent parotitis; RIS: radioiodine-induced sialadenitis; SS: Sjögren’s syndrome; and ES: eosinophilic sialodochitis.

**Figure 3 jcm-14-07560-f003:**
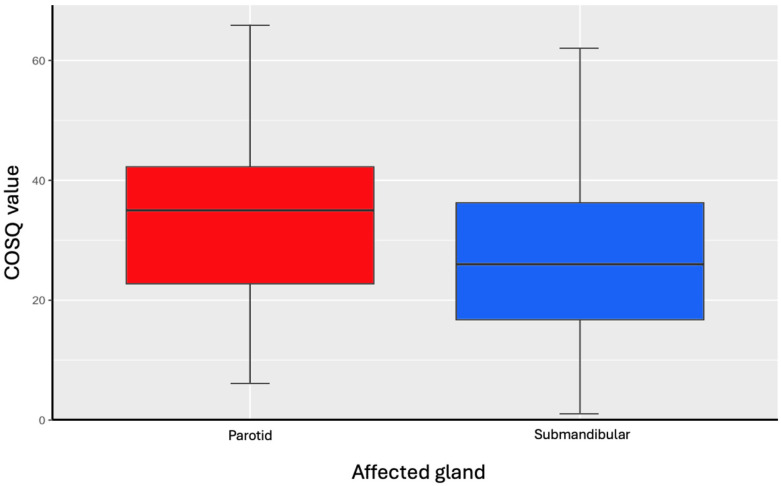
COSQ values in relation to affected salivary gland.

**Figure 4 jcm-14-07560-f004:**
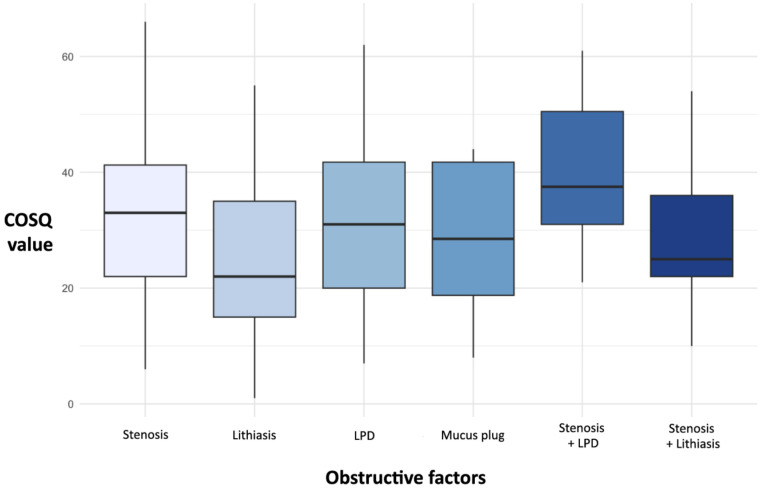
Box and whisker plot of COSQ values by obstructive causes (>1% of the sample).

**Table 1 jcm-14-07560-t001:** Epidemiological characteristics of the study population.

Epidemiological Data	Value
**Sex** (% [n])	
Male	28.05% [n = 78]
Female	71.94% [n = 200]
Female-to-male ratio	2.56:1
**Age** (years) [mean ± SD]	49.67 ± 14.12
**Median duration of symptoms** (median [range])	2.15 [0.2–9]
**Affected salivary gland** (% [n])	
Parotid	50% [n = 172]
Submandibular	50% [n = 172]

**Table 2 jcm-14-07560-t002:** COSQ scores according to epidemiological variables and their corresponding statistical comparisons.

Epidemiological Factor	COSQ Value (Median [Range])	*p*-Value
Age		0.71
Years of evolution		0.01297
Sex		
Male	20 [5–59]	<0.005
Female	33 [1–66]	
Affected gland		
Submandibular	26 [1–62]	
Parotid	35 [6–66]	<0.005

**Table 3 jcm-14-07560-t003:** Distribution of obstructive factors, dilation, and potentially obstructive entities (POEs), along with their associated COSQ scores and statistical comparisons.

Obstructive Cause	% (n)	Median COSQ [Range]	*p*-Value
Stenosis	41.86% (144)	33 [6–66]	**<0.005**
Lithiasis	29.36% (101)	22 [1–55]	
LPD ^1^	15.12% (52)	31 [7–62]	
Mucus plug	6.39% (22)	28.5 [8–44]	
Stenosis + LPD	3.48% (12)	37.5 [21–61]	
Stenosis + Lithiasis	2.62% (9)	25 [10–54]	
Stenosis + Mucus plug	0.58% (2)	59 [59–59]	
LPD + Mucus plug	0.29% (1)	34 [34–34]	
Lithiasis + LPD	0.29% (1)	16 [16–16]	
**Grade of stenosis**			**<0.005**
S1	1.21% (2)	17 [13–21]	
S2	49.69% (82)	33.5 [6–63]	
S3	20.6% (34)	29 [10–59]	
S4	20% (33)	40 [13–66]	
Combined	8.48% (14)	31.5 [6–55]	
**Grade of dilation**			
D1	48.39% (15)	33 [8–45]	
D2	22.58% (7)	33 [21–60]	
D3	29.03% (9)	35 [13–60]	0.4928
**Lithiasis per location**			
Ductal	72.97% (81)	22 [1–55]	
Hilar	27.02% (30)	20 [5–43]	0.8602
**Potentially Obstructive Entities (POEs)**			
RIS ^2^	9.3% (32)	36 [22–63]	
SS ^3^	6.69% (23)	42 [21–63]	
ES ^4^	0.58% (2)	65 [64–66]	
JRP ^5^	0.29% (1)	33 [33–33]	**0.0312**

^1^ LPD: Lack of Papilla Distensibility. ^2^ RIS: RadioIodine Sialadenitis. ^3^ SS: Sjögren Syndrome. ^4^ ES: Eosinophilic Sialodochitis. ^5^ JRP: Juvenile Recurrent Parotitis.

**Table 4 jcm-14-07560-t004:** Distribution of obstructive factors and potentially obstructive entities (POEs) per affected salivary gland, with associated COSQ scores and statistical comparisons.

Obstructive Cause per Gland	% (n)	*p*-Value	Median COSQ [Range]	*p*-Value
**Stenosis**				
Parotid	75.76% (125)		35 [6–66]	
Submandibular	24.24% (40)	**<0.005**	27.5 [8–61]	0.105
**Lithiasis**				
Parotid	24.32% (27)		27 [8–44]	
Submandibular	75.68% (84)	**<0.005**	18.5 [1–55]	0.064
**LPD** ^1^				
Parotid	17.65% (12)		38.5 [20–55]	
Submandibular	82.53% (56)	**<0.005**	30.5 [7–62]	**0.0164**
**Mucus plug**				
Parotid	84% (21)		40.0 [8–59]	
Submandibular	16% (4)	**<0.001**	20.0 [8–31]	0.0877
**Dilation**				
Parotid	77.42% (24)		32.5 [8–60]	
Submandibular	22.58% (7)	**<0.005**	41 [13–48]	0.2016
**Potentially Obstructive Entities (POEs)**				
RIS ^2^				
Parotid	75% (24)		36.5 [22–63]	
Submandibular	25% (8)	**0.0053**	27 [22–47]	0.073
SS ^3^				
Parotid	88% (22)		41 [21–63]	
Submandibular	22% (3)	**<0.005**	45 [34–55]	0.6153
ES ^4^				
Parotid	100% (2)		65 [64–66]	
Submandibular	0% (0)	0.498	-	-
JRP ^5^				
Parotid	100% (1)		33 [33–33]	
Submandibular	0% (0)	1	-	-

^1^ LPD: Lack of papilla distensibility. ^2^ RIS: Radioiodine Sialadenitis. ^3^ SS: Sjögren Syndrome. ^4^ ES: Eosinophilic Sialodochitis. ^5^ JRP: Juvenile Recurrent Parotitis.

## Data Availability

The original contributions presented in this study are included in the Appendix A archive. Further inquiries can be directed to the corresponding author.

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
