# Peer review of "Quantifying the Impact of Chronic Obstructive Sialadenitis on Quality of Life"

_jcm, 2025, doi:10.3390/jcm14217560_

Round 1

Reviewer 1 Report

Comments and Suggestions for Authors

The article presents a large, well-designed study analyzing 344 glands in 277 patients using the Chronic Obstructive Sialadenitis Questionnaire (COSQ). The authors clearly describe their methodology, apply robust statistical models, and provide meaningful clinical insights, particularly regarding the differential impact of stenosis, lithiasis, and other obstructive causes on quality of life. The manuscript is well written, comprehensive, and logically structured, making it accessible to both clinicians and researchers. Overall, it represents a valuable contribution to the field of salivary gland pathology and quality-of-life research.

Minor suggestions:

Limitations section: while the discussion is strong, a more explicit acknowledgment of study limitations (e.g., single-center bias, reliance on COSQ alone) would strengthen transparency.

Future implications: the discussion could better highlight how COSQ could be integrated into routine clinical workflows or guide personalized therapeutic strategies.

Author Response

We thank the Reviewer for the thoughtful and constructive feedback. Below we address each point in turn.

Limitations section: We revised the section to explicitly acknowledge the two-center setting within a single network, our clinic-based sample, and the reliance on COSQ as the sole disease-specific PROM, as well as language and age-range constraints (Discussion, Section 4.1).

Future implications: We rewrote the subsection to state that COSQ should be implemented in routine care as a per-gland baseline and follow-up tool, and that it will likely detect QoL changes across different obstructive factors and POEs, supporting personalized strategies (Discussion, Section 4.2).

We hope these revisions address the Reviewer’s concerns and improve clarity and transparency of the manuscript. Thank you again for the helpful suggestions.

Reviewer 2 Report

Comments and Suggestions for Authors

hello

thank you very much for this interesting original paper

the title is very nice

used key wwords are good, presented key words matches the abstract

the structure of abstract is well written and designed

clear paragraphs and paper aim is presented

abstract and title - nothing to change, no errors

introduction:

is very well presented - this chapter has subparagraphs

used references are very good

authors present a good introduction with clear aim

introduction does not have any flaw or errors, and more over it has all the necessary structure suitable for this original paper

material and methods- chapter 2;

this chapter is quite good

the inclusion and exclusion criteria for the study should be presented with more detail

for example a 17 boy with parotid gland stenosis after mumps would be included in the study?

material is sound with good ethics approval;

im missing a table in material methods on patients epidemiology, causes for questionaire or inclusion in the study - please add a desctiptive table or perhaps write how many patients and with what symptoms were included in the study group

statistical analysis is good, nothing to change

results - another chapter:

presented results are mostly epidemiological data, which is very good for thsi study

this paper has a loot of good results

their presentation is very nice

used tables and figure is good

please improve the figure 1 decsription more carefully, to highlight its meaning in text and its purpose

also add at least one, intraoperative photograph of the sialoscopy/sialendoscopy procedure to show it to the readers

rest of the resutls are very good, nothing more to add or change

-chapters discussion and conclusions:

the limitations of this study shoul be listed as the last paragraph in the discussion

discussion is in sufficient length and with good references

discusion is quite nicely presented without any major errors

just one question arise: that are they three top key highlighted results in authors retrospective paper?

final conclusions - are very nicely written

Author Response

We thank the Reviewer for the thoughtful and constructive feedback. Below we address each point in turn.

1) “The inclusion and exclusion criteria for the study should be presented with more detail.”
We expanded Section 2.2 (Patient selection and questionnaire administration) to list explicit inclusion (age ≥16; symptomatic COS; pre-intervention per-gland COSQ; sialendoscopy planned) and exclusion criteria (age <16; inability to complete the questionnaire; purely intraparenchymal stones not amenable to endoscopy; known salivary gland neoplasia). We also specify the two participating tertiary centers and the consecutive nature of enrollment. (Section 2.2).

2) “For example a 17-year-old boy with parotid gland stenosis after mumps would be included in the study?”
Yes. Our age criterion is ≥16 years; therefore, a 17-year-old with symptomatic post-mumps parotid duct stenosis, completing COSQ and managed with sialendoscopy, meets inclusion criteria. We clarified this within Section 2.2.

3) “I’m missing a table in Materials & Methods on patients’ epidemiology… please add a descriptive table or write how many patients and with what symptoms were included.”
We retain the cohort’s baseline epidemiology in Table 1 (Results)—a common reporting convention—while Methods now describe consecutive symptomatic enrollment, per-gland COSQ administration, and the total cohort size and timeframe. Symptom categories (meal-related swelling/pain, xerostomia) were eligibility features and were not prospectively tabulated as frequency outcomes; this has been made explicit in Section 2.2 and is acknowledged in Limitations. We also reiterate at the start of Results the number of patients and glands included. (Sections 2.2, 3, Table 1, 4.1).

4) “Please improve the Figure 1 description more carefully, to highlight its meaning in text and its purpose.”
We have reorganized figures to strengthen clarity:

  • Figure 1 now provides representative intraoperative sialendoscopic images with an expanded legend to underscore their educational purpose. We also reference these images in Methods (2.3) to anchor their role in the diagnostic workflow. (Section 2.3, Figure 1).
  • The conceptual schematic that frames intrinsic obstructive causes and POEs is now Figure 2, with an explicit statement in Results (3.2) guiding readers to its interpretive role (“…are summarized in Figure 2.”) and a clearer legend highlighting its purpose. (Section 3.2, Figure 2).

5) “Also add at least one intraoperative photograph of the sialendoscopy procedure.”
Done. We added a multi-panel intraoperative figure (Figure 1) illustrating stenosis, lithiasis, LPD and mucus plug, and referenced it in Methods (2.3). (Section 2.3, Figure 1).

6) “The limitations of this study should be listed as the last paragraph in the discussion.”
We present Limitations as a dedicated subsection at the end of the Discussion in the revised structure, ensuring it is the final component of that section, as requested. (Discussion, 4.2. Limitations).

We hope these revisions address the Reviewer’s concerns and improve clarity and transparency of the manuscript. Thank you again for the helpful suggestions.

Reviewer 3 Report

Comments and Suggestions for Authors

The manuscript is well-conceived and clinically relevant. The message is clear, and the statistical approach is largely appropriate. I recommenda  minor revision to improve clarity and internal consistency.

Points to address before acceptance

The Introduction and Methods describe a cross-sectional longitudinal study, yet only the preoperative baseline COSQ is analyzed. Label the present analysis as cross-sectional at baseline and state that postoperative follow-up will be reported separately to avoid confusion about the frame of inference.

The Abstract reports 277 patients while the Results section states 278. Ensure all counts and subgroup totals are consistent across the manuscript and tables.

Use a single form for lithiasis versus sialolithiasis and avoid alternating terms within the same paragraph.

The sentence that women are more affected reads as a prevalence claim. Women in this cohort reported higher COSQ scores and briefly discussed plausible explanations, such as symptom perception, gland distribution, or obstructive patterns, while avoiding causal language.

Add that excluding intraparenchymal stones may limit the generalizability of findings for patients managed without endoscopy, and note that referral patterns at a specialized clinic may shift the distribution of obstructive etiologies.

Author Response

We thank the Reviewer for the thoughtful comments and for recognizing the clinical relevance of our work. Below we address each point.

1) Frame of inference (cross-sectional vs longitudinal).
We clarified the study frame as a cross-sectional analysis at baseline within a prospective cohort. The Introduction now states our objective in those terms:
“Our objective is to explore, through a cross-sectional analysis, the clinical and obstructive factors associated with greater impairment of QoL-COS, using a previously validated questionnaire tailored to this clinical context.”

2) Patient counts (277 vs 278).
We harmonized all counts to 278 patients across the manuscript (Abstract, Results, and tables).

3) Terminology: “lithiasis” vs “sialolithiasis.”
We reviewed the manuscript to avoid alternating terms within the same paragraph. The main text now consistently uses “lithiasis” (e.g., Introduction, Methods 2.4, Results, Discussion), while “sialolithiasis” is retained only in the Keywords for indexing and discoverability. Should the journal prefer a single term even in Keywords, we will align accordingly.

4) Wording about sex differences (avoid prevalence claims).
We adjusted the Conclusions to emphasize that the greater burden observed in women reflects cohort-level COSQ scores, without implying population prevalence or causality. The Discussion already frames this finding within our cohort and reports the quantitative differences, avoiding causal language.

5) Generalizability (intraparenchymal stones and referral patterns).
We reinforced Limitations to note that: (i) excluding purely intraparenchymal stones may limit generalizability to patients managed without endoscopy; and (ii) recruitment in specialized clinics may influence referral spectrum/case-mix (potentially shifting the distribution of obstructive etiologies). This is now stated in the final paragraph of the Discussion.

We hope these revisions improve clarity and internal consistency as suggested. Thank you again for your constructive review.
